# Spacer Block Technique Was Superior to Intramedullary Guide Technique in Coronal Alignment of Femoral Component after Fixed-Bearing Medial Unicompartmental Knee Arthroplasty: A Case–Control Study

**DOI:** 10.3390/medicina59010089

**Published:** 2022-12-31

**Authors:** O-Sung Lee, Myung Chul Lee, Chung Yeob Shin, Hyuk-Soo Han

**Affiliations:** 1Department of Orthopedic Surgery, Eulji University School of Medicine, Uijeongbu-si 11759, Republic of Korea; 2Department of Orthopedic Surgery, Seoul National University College of Medicine, Seoul 03080, Republic of Korea

**Keywords:** spacer block, intramedullary rod, femorotibial congruence, unicompartmental arthroplasty

## Abstract

*Backgrounds and Objectives*: The spacer block technique in unicompartmental knee arthroplasty (UKA) has still a concern related to the precise position of the component in the coronal and sagittal planes compared to intramedullary guide technique. The purposes of this study were to explore whether the spacer block technique would improve the radiological alignment of implants and clinical outcomes compared with the outcomes of the intramedullary guide technique in fixed-bearing medial UKA. *Materials and Methods*: In total, 115 patients who underwent unilateral, fixed-bearing medial UKA were retrospectively reviewed and divided into group IM (intramedullary guides; n = 39) and group SB (spacer blocks; n = 76). Clinical assessment included range-of-motion and patient-reported outcomes. Radiological assessment included the mechanical femorotibial angle, coronal and sagittal alignments of the femoral and tibial components, and coronal femorotibial congruence angle. *Results*: All clinical outcomes showed no significant differences between groups. The coronal femoral component angle was valgus 2.4° ± 4.9° in IM group and varus 1.1° ± 3.2° (*p* < 0.001). In group IM, the number of outlier in coronal femoral component angle (<−10° or 10°<) was 3 cases, while in group SB, there was no outlier (*p* = 0.014). The coronal femorotibial congruence angle was significantly less in group SB (mean 1.9°, range, −3.2°~8.2°) than in group IM (mean 3.4°, range, −9.6°~16.5°) (*p* = 0.028). *Conclusions*: In the group SB, the coronal alignment of femoral component was closer to neutral, and outlier was less frequent than in the group IM. The spacer block technique was more beneficial in achieving proper coronal alignment of the femoral component and congruence of femorotibial components compared to the intramedullary guide technique in fixed-bearing medial UKAs.

## 1. Introduction

Unicompartmental knee arthroplasty (UKA) is a more attractive treatment option than total knee arthroplasty (TKA) for patients with unicompartmental knee osteoarthritis; it is associated with less postoperative morbidity, more rapid recovery, more physiological kinematics, and greater patient satisfaction [1,2,3,4]. However, the national registries of Europe and Oceania consistently report inferior long-term survivorship of UKAs compared to TKAs [5]. It has been suggested that both implantation and ligament balancing during UKA may be rather inaccurate. Inappropriate component alignment may trigger edge loading, early polyethylene wear, and a high revision rate [6,7]. Mobile-bearing UKA features a round-on-round interface and provided that the orientation of knee components is within a tolerable range, any effect on knee mechanics may be negligible [8,9]. However, fixed-bearing UKA features a round-on-flat interface; thus, it exhibits relatively narrower tolerance in terms of component-to-component malpositioning [10]. These characteristics of fixed-bearing UKA are associated with risks of edge loading and rapid wear [11,12].

Although new patient-specific instrument and navigation techniques are under development, conventional UKA techniques using intramedullary guides or spacer blocks remain widely used. In the context of mobile-bearing UKA, it remains unclear whether new intramedullary rod guides improve radiological alignment and clinical outcomes [13,14]. During fixed-bearing UKA, intramedullary rods have been widely used to align the femoral component. However, given the unsatisfactory accuracy of intramedullary alignment, extramedullary alignment using a spacer block or a tensor device is becoming popular [15,16,17,18]. Gap creation using a spacer block aligns the distal femoral resection parallel to the tibial resection in extension; the component gap can be predicted prior to femoral osteotomy. However, it is difficult to define a gap that reflects the physiologically relevant soft tissue balance; a surgeon’s technical proficiency and subjective assessment of gap balancing are contributing factors.

In contrast to a TKA, coronal alignment of the components of a UKA is independent of the axis of the lower leg. Rather, the femoral and tibial component have an effect on each other especially in their coronal positioning in knee extension [19]. Although the spacer block technique shows promise in terms of reconstructing physiological joint tension, limited data are available regarding optimal femoral component positions in the coronal and sagittal planes. This technique produces the alignment of the femoral component according to positioning of the spacer introduced after cutting the tibia. Surgeon connects the block to this spacer for cutting the distal femur parallel to the tibial cutting surface on the coronal plane.

The purposes of the present study were to explore whether the spacer block technique improved implant radiological alignment and clinical outcomes, compared to use of an intramedullary guide, in patients undergoing fixed-bearing medial UKAs. It was hypothesized that the use of a spacer block would improve postoperative femoral component positioning and the congruence of femorotibial (FT) components.

## 2. Materials and Methods

In total, 129 consecutive patients who underwent unilateral, fixed-bearing medial UKA from February 2010 to May 2018 were eligible for inclusion. Fourteen were excluded for the following reasons: fewer than 2 years of follow-up (n = 4), postoperative complications including an infection (n = 1) and loosening (n = 3), and missing clinical data or radiographs (n = 6). Finally, 115 patients who were available for clinical and radiographic evaluation with minimum 2-year follow-up were divided into two groups according to the surgical instrument and implant used: group IM (intramedullary rod guide; n = 39) (MIS Miller/Galante, Zimmer, Warsaw, IN, USA) and group SB (spacer block; n = 76) (Sigma High Performance Partial Knee, DePuy Synthes, Warsaw, IN, USA). The study was approved by the institutional review board (approval no. H-2005-180-1126). The indications for UKA were identical in both groups: A clinical and radiographic diagnosis of isolated medial compartmental osteoarthritis, passive knee flexion contracture <10°, active maximal flexion >100°, and varus deformity <10°. Patient demographics were collected from medical records; these data are summarized in Table 1. There were no significant differences in demographic characteristics between the groups.

## 3. Surgical Technique

All patients were operated upon by two experienced surgeons (HSH and MCL). A medial mini-parapatellar arthrotomy was performed through a skin incision of approximately 10 cm in length. After osteophyte resection, medial soft tissue release was confined to the deep medial collateral ligament. No further release to correct varus alignment was performed. The tibia was first cut using extramedullary guides to ensure coronal alignment between the native medial proximal tibial angle (MPTA) and 0° varus/valgus to the mechanical tibial axis. The sagittal alignment was aimed to be similar to the native medial posterior tibial slope. For femoral preparation, in group IM, a hole was created for insertion of the intramedullary guide (using an 8 mm drill), 1 cm anterior to the origin of the posterior cruciate ligament and immediately anterior to the femoral intercondylar notch. The intramedullary rod was inserted, and a distal femoral resection guide was then attached with the valgus angle measured on preoperative standing whole lower extremity radiographs. In group SB, a spacer block of the appropriate size (thickness) was placed; the gap from full extension to full flexion was checked. The goal was to ensure that the flexion and extension gaps were balanced with the spacer blocks in various sizes. A distal femoral cutting guide of the same thickness was inserted and pinned in place with the knee in extension. In both groups, femoral component rotation was performed to create a rectangular flexion gap. For confirmation of proper gap, the flexion gap is measured in about 100° flexion using the gap checking device that matches the spacer block thickness used for cutting the distal femur. Additionally, then, the extension gap is measured with this device. The correct gap indicating natural tension was assessed subjectively by checking whether the gap was loose with a thinner gap checker and tight with a thicker one (Figure 1). All components were cemented in both groups. All patients followed the same postoperative rehabilitation protocol, and full weight-bearing (as tolerated) the day after surgery.

## 4. Clinical and Radiological Assessment

Clinical and radiological evaluations were performed preoperatively, at 6 weeks and 3 months postoperatively, and annually thereafter. Information was retrieved from the clinical data repository regarding the pre- and post-operative ranges of motion, as well as clinical scores including the Hospital for Special Surgery score, the Knee Society Knee and Function Scores, the pain visual analog scale score, and the Western Ontario McMaster Universities Osteoarthritis Index.

Standardized anteroposterior and lateral radiographs of the knee, whole lower extremity standing anteroposterior (AP) radiographs were obtained for radiologic assessment. All radiological parameters were measured twice at a 2-week interval using a Picture Archiving and Communication System by two orthopedic surgeons who were not members of the surgical team. The preoperative mechanical FT varus angle, MPTA, and posterior tibial slope angle were measured. Postoperatively, the mechanical FT varus angle, coronal femoral component (α) angle and coronal tibial component (β) were determined by measuring varus (−)/valgus (+) alignment of each component relative to the long axis of the tibia on AP radiograph [11,20]. Sagittal femoral component angle was determined as the angle between a line perpendicular to the component part placed on distal femur cut and a long axis of the femur [21]. Sagittal tibial component angle was determined as the angle between a line perpendicular to the long axis of the tibia and a line tangential to the tibial component on a lateral radiograph (Figure 2). The coronal FT component congruence angle was defined as the angle of tilt between the line perpendicular to the long axis of the femoral component surface and the clearly visible lower margin of the tibial component in the AP view; lateral convergences between these two lines were assigned positive values. Clinical and radiological results achieved at 2 years postoperatively were used for comparison between the groups.

## 5. Statistical Analysis

Continuous variables are presented as means with standard deviations; categorical variables are presented as frequencies with percentages. The Kolmogorov–Smirnov test was used to explore the normality of data distribution. Comparisons were performed using Pearson chi-square test and Student’s *t*-test. Intraclass correlation coefficients were calculated to assess the inter- and intraobserver reliabilities of radiological measurements. To reveal the sample size of this study sufficient for adequate power, the statistical software G*Power (Erdfelder, Faul, Lang and Buchner, 2014) was used for power analysis. A post hoc power analysis was performed and showed a sufficient sample size for this retrospective study with an alpha of 0.05 and a power > 0.8.) All statistical analyses were performed using SPSS software (ver. 25.0.0). A *p*-value < 0.05 was considered to indicate statistical significance.

## 6. Results

Neither demographic nor preoperative clinical and radiological data differed significantly between the two groups, with the exception of the mechanical FT varus angle (group IM, 6.6° vs. group SB 5.3°; *p* = 0.05) (Table 1). No intraoperative complications or instrument-related problems were encountered in either group. The postoperative clinical and radiological data of both groups are summarized in Table 2. The clinical outcomes (except the maximal knee flexion angle) improved significantly in both groups. No significant differences were found in any postoperative clinical parameter in either group.

However, several radiological parameters differed significantly between the groups. Although the extent of correction of the mechanical FT angle did not differ between the groups, the postoperative mechanical FT varus angle was smaller in group SB than in group IM (−1.8° ± 3.7° vs. −0.3° ± 3.2°, *p* = 0.025), which was similar to the preoperative difference. The coronal femoral component angle (α) was valgus 2.4° ± 4.9° in group IM, and varus 1.1° ± 3.2° in group SB, respectively (*p* < 0.001) (Figure 3).

The number of outlier in α angle (<−10 or 10<) was 3 in group IM, and 0 in group SB, respectively (*p* = 0.014) [22]. The postoperative coronal tibial component angle (β) significantly differed between the two groups (group IM, varus 1° ± 4.3° vs. group SB, 3.0° ± 2.4°; *p* = 0.001), and MPTA change also significantly differed between the groups (group IM, 2.4° ± 4.9° vs. group SB, 0.2° ± 2.8°; *p* = 0.003). Sagittal femoral and tibial component angles showed no significant difference between the two groups. Finally, the coronal FT component congruence angle of group SB (mean, 1.9°; range, −3.2 to 8.2°) was less than that of group IM (mean, 3.4°; range, −9.6 to 16.5°) (*p* = 0.028) (Figure 4). All radiological measurements exhibited excellent intraclass correlation coefficients in terms of both intra- and interobserver reliabilities (0.81–0.92).

## 7. Discussion

The principle findings of the present study were as follows: (1) the femoral component implanted by the spacer block technique has more neutral alignment relative to the long axis of the tibia with less outliers (>±10° deviation from neutral) [22]. (2) the mean and variation of the coronal FT component congruence angle in patients for whom the spacer block technique was used were lower than the corresponding values in patients treated using the intramedullary guide technique after fixed-bearing medial UKA. However, there were no significant between-group differences in clinical outcomes on short-term follow-up.

The success of UKA relies on appropriate implant alignment and positioning, as well as correct soft tissue tensioning, to ensure a balanced flexion-extension gap and stability [1,2,3,4]. Good surgical technique and instruments that avoid edge loading attributable to component-to-component mal-positioning are essential. In contrast to TKA, coronal alignment of the femoral component during UKA does not affect the leg axis; however, it influences relative component positioning in extension [8]. During UKA, the appropriateness of intramedullary alignment of the femoral component has been questioned, given the limited precision of manipulation, as well as potential morbidity created by trochlear cartilage perforation and medullary canal opening [15,16,17]. The spacer block technique seeks to ensure a rectangular, medial extension gap, thereby achieving maximum possible contact between femoral and tibial components. In this technique, the femoral component is aligned with the tibial cut by introduction of a spacer in extension after cutting the tibia. The intraoperatively, symmetrically resected medial extension gap also ensures parallel positioning of implants in the coronal plane.

In terms of evaluation of postoperative implant alignment after TKA, many studies have indicated that proper alignment of implant has a beneficial impact on implant survival and functional outcomes after TKA [23,24,25]. However, in contrast of TKA, the impact of implant alignment on long-term survival and clinical outcomes after UKA is unclear. UKA is known to have a widely acceptable range of component alignment within ±10° of coronal and sagittal alignment for the femoral component and ±5° coronal and sagittal alignment for the tibial component [11,22]. Despite acceptable deviation of postoperative limb alignment from neutral alignment may not have an effect on outcomes, far outliers of deviation may lead to harmful effect on survival and clinical outcomes after UKA. In our study, although the mean values of the coronal femoral component angle in both groups were within varus/valgus 3 degrees, the femoral component in the SB group has more neutral alignment relative to the long axis of the tibia with less outliers (>±10° deviation from neutral). Additionally, the mean coronal FT component congruence angle in SB group was lower than that in IM group, which may be associated with a risk of accelerated wear in the components.

Studies of mobile-bearing UKA using spacer blocks demonstrated a range of femoral component coronal alignments (8–11° valgus and 8–13° varus) [8,11]. However, during mobile-bearing UKA, because of the spherical nature of the femoral component, femoral malalignment of 10° and tibial malalignment of 5° did not hinder good clinical outcomes [11]. One retrospective study evaluated 193 consecutive patients who had undergone fixed-bearing medial UKA using the spacer technique [19]. The clinical results were comparable to those afforded by the intramedullary guide technique. However, the precision was lower and outliers were more frequent. In the present study, femoral implants exhibited no differences in terms of coronal or sagittal plane accuracies between groups SB and IM; however, the component-to-component congruence was better in group SB. When using the spacer block technique, the position of the femoral component is determined in the planes of the tibial component. Because femoral alignment is thus dependent on tibial alignment, the femoral and tibial components lie parallel, with only 1.9 ± 2.6° of asymmetry. This alignment is better than that of the intramedullary guide technique, where the femoral component position is independent of tibial component position. Given the narrower tolerable congruence between the femoral and tibial components of fixed-bearing UKA, compared to mobile-bearing UKA, the spacer block technique may be superior to the intramedullary guide technique.

When using the spacer block technique, care must be taken to avoid transference of any tibial malalignment to the femoral component; tibial resection must be very precise. However, no consensus has emerged regarding ideal positioning of the tibial component during UKA. In the present study, we first cut the tibia using extramedullary guides to achieve coronal alignment between the native MPTA and the 0° varus/valgus mechanical tibial axis; we also achieved sagittal alignment similar to that of the native medial posterior tibial slope. The mean coronal tibial component angle differed by 2° between groups IM and SB. Although the angle of group IM was more neutrally aligned to the mechanical tibial axis, compared to the angle of group SB (89° and 87°, respectively), the short-term clinical outcomes did not differ between the two groups. The significance of coronal tibial or femoral component alignment, and congruence between femoral and tibial components, should be explored over a longer follow-up period. Importantly, there is a need to address wear, aseptic loosening, and implant survival.

The optimal knee alignment during UKA remains controversial; few reports have addressed this aspect. The ideal component alignments and their effects on long-term UKA outcomes remain unknown. Overcorrection of a tibiofemoral deformity should be avoided to reduce the risk of degenerative change in the contralateral compartment; many authors have recommended that knee alignment should be relatively undercorrected during medial UKA [17,26]. However, undercorrection would increase the load on the medial compartment, which may accelerate polyethylene wear. In a retrospective case series (471 UKAs), more than 50% of failures occurred within 5 years of implantation [26]. The major cause of failure was development of other-compartment arthritis (39.5%), followed by aseptic loosening (25.4%). The FT angles tended to be greater in patients with contralateral arthritis. However, pre-medial UKA radiographs were not available for most patients. The use of a spacer block to determine femoral component position during UKA is associated with a risk of overcorrection [27]. However, the intramedullary guide technique is associated with risks of overstuffing and overcorrection when determining final polyethylene thickness. In the present study, we achieved a mean of 5° of correction and slight varus alignment in both groups. The correction was similar to that other studies (4° to 5°) [19,28].

The present study had several limitations. First, it was retrospective in nature and did not involve randomization; selection bias may have influenced the findings. The preoperative mechanical FT angles significantly differed between the two groups. Second, we compared two different kinds of implants; this might have caused other forms of undetected bias. However, in terms of evaluation of the coronal and sagittal alignments of the tibial component, it is reasonable that our measurement method provides constant results regardless of the design of the implant (Figure 1). Additionally, in terms of measurement of the sagittal alignment of femoral component, we used the distal femur cutting line in order to minimize the bias due to the difference of the implants. Additionally, in terms of the coronal alignment of femoral component, we tried to minimize the bias by using the measuring method introduced by Gulati et al. [11], which is based on long axis of the femoral component regardless of the type of implant and most widely used in the evaluation of component alignment after UKA. Third, postoperative coronal tibial component angle (β) significantly differed between the SB and IM groups, which could affect the coronal femoral component angle (α). However, the more varus the coronal tibial component angle was set, the larger the coronal FT component congruence angle would become in the IM group (Figure 5). Therefore, we could reduce the concern about errors caused by the difference in the coronal tibial component angle after surgery. Finally, all measurements were performed using simple radiographs alone. Accordingly, component rotational errors could not be assessed.

## 8. Conclusions

The spacer block technique was more beneficial in achieving proper coronal alignment of the femoral component compared to the intramedullary guide technique in fixed-bearing medial UKAs. The coronal alignment of femoral component was closer to neutral, and outlier was less frequent. Moreover, the congruence of FT components afforded by the spacer block technique was superior to that of the intramedullary guide technique after fixed-bearing medial UKA. Bone resection using a spacer block ensured that the implants were parallel in the coronal plane and may reduce edge loading. However, there were no significant differences in clinical outcomes between the two techniques on short-term follow-up.

## Figures and Tables

**Figure 1 medicina-59-00089-f001:**
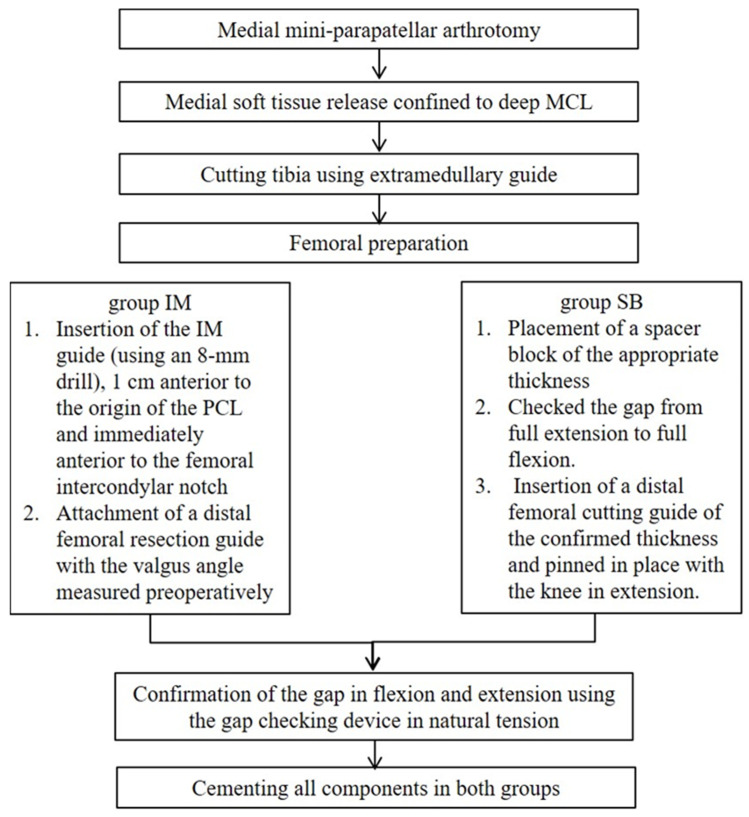
Schematic diagram of surgical procedure in UKA. MCL; medial collateral ligament, IM; intramedullary, SB; space block.

**Figure 2 medicina-59-00089-f002:**
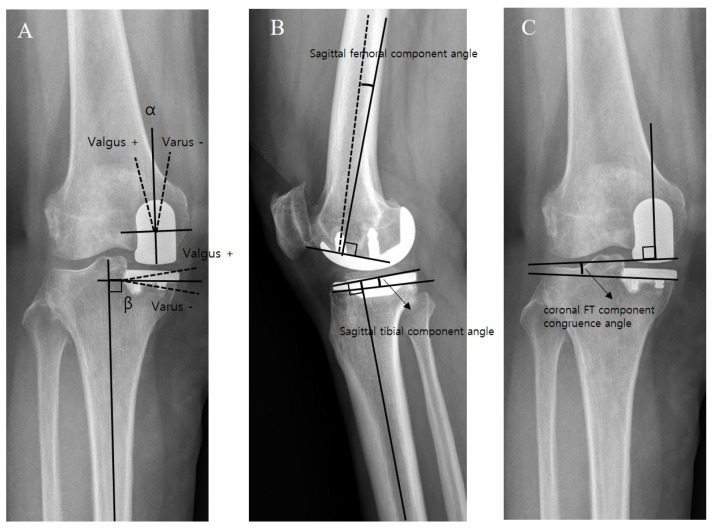
Measurement of alignments of the femoral and tibial components on postoperative radiographs. (**A**) Coronal femoral component angle (α) and coronal tibial component angle (β). (**B**) Sagittal femoral component angle and sagittal tibial component angle. (**C**) Coronal FT component congruence angle.

**Figure 3 medicina-59-00089-f003:**
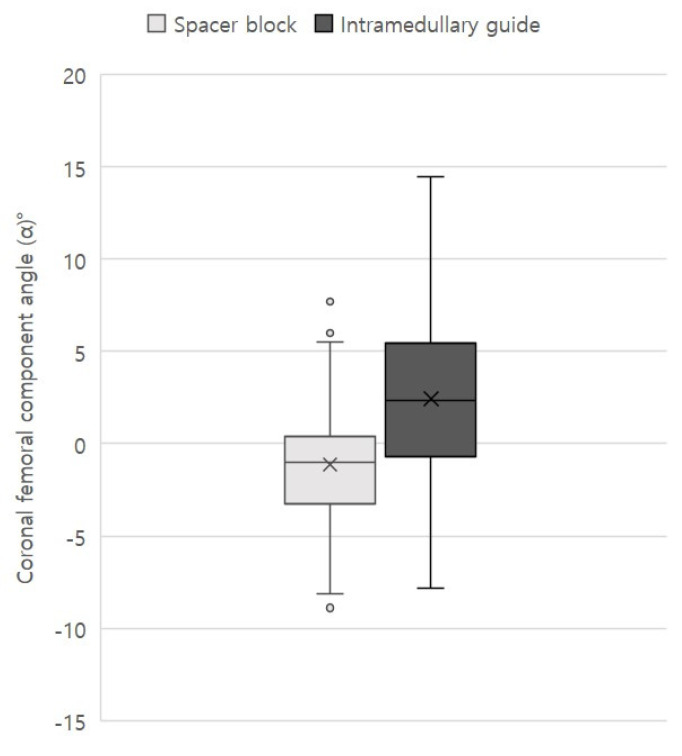
Boxplot of the distribution of the coronal femoral component angle in group SB and IM.

**Figure 4 medicina-59-00089-f004:**
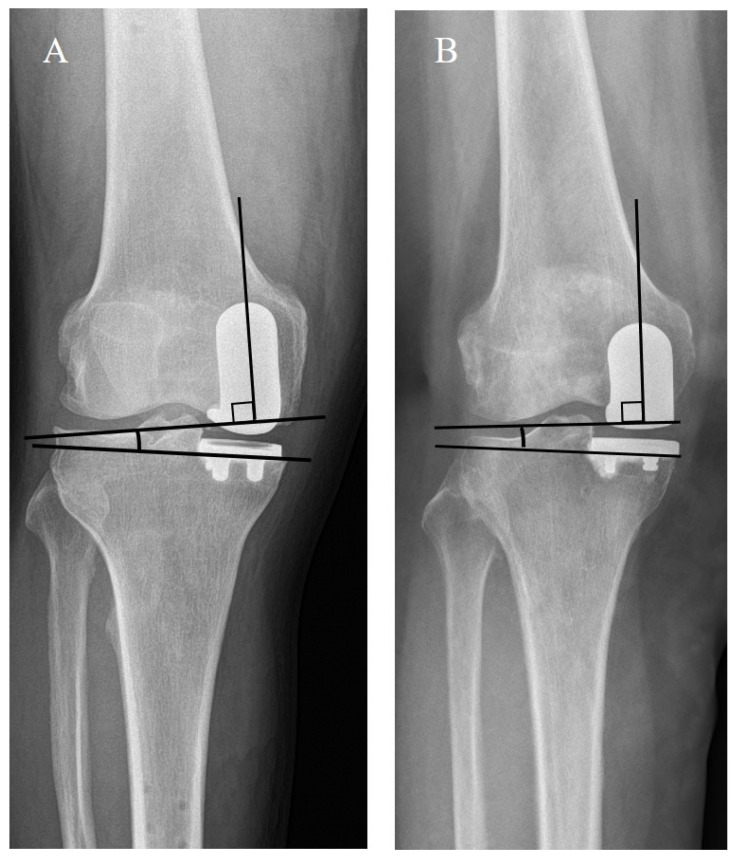
Radiographs showing that the coronal FT component congruence angle of group SB is smaller than that of group IM. (**A**). Showing the coronal FT component congruence angle of group IM, (**B**) Showing the coronal FT component congruence angle of group SB.

**Figure 5 medicina-59-00089-f005:**
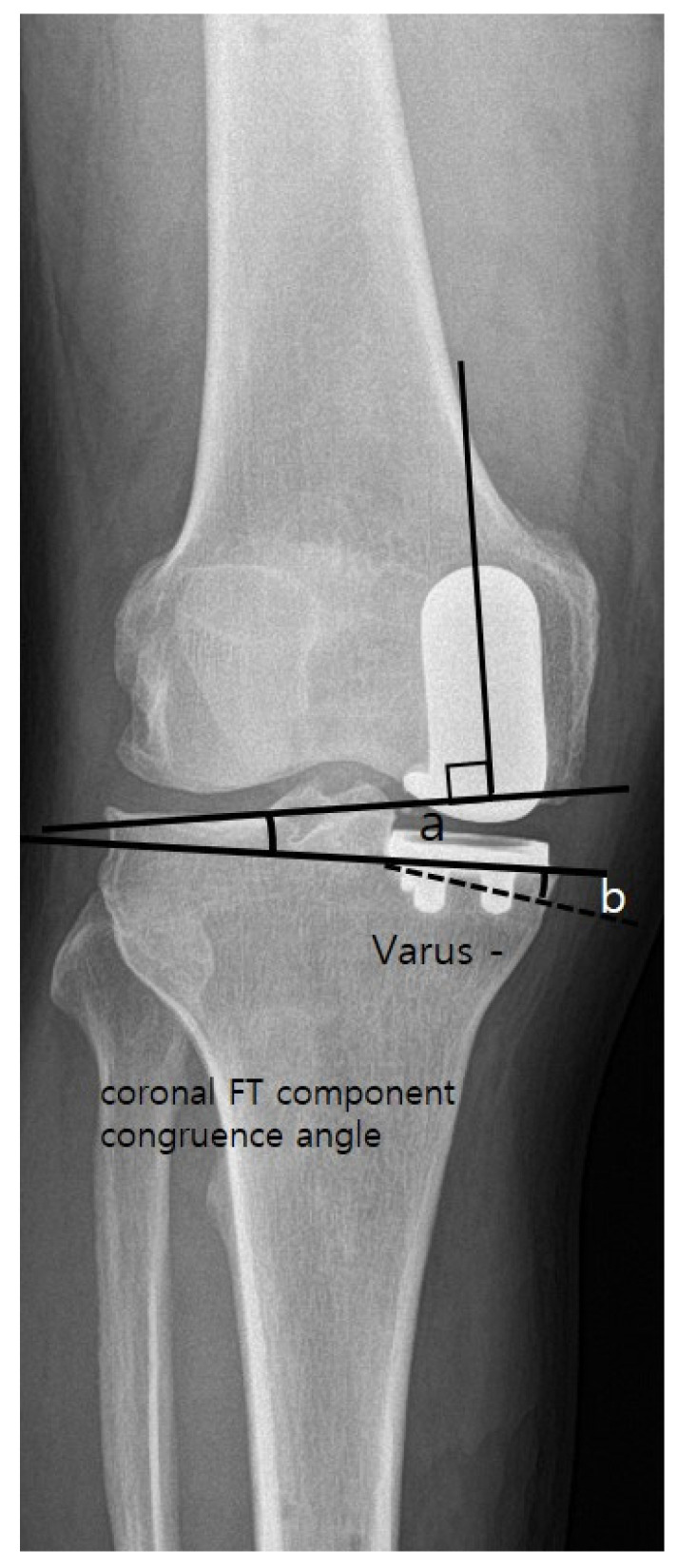
Figure that expresses the imaginary lines showing the coronal FT component congruence angle increases as the varus degree of coronal tibial component angle increases. The (**a**) indicates the coronal FT component congruence, and the (**b**) indicates the the varus degree of coronal tibial component angle.

**Table 1 medicina-59-00089-t001:** Demographic and preoperative clinical and radiological data of patients in the intramedullary guide and spacer block groups.

Variable	Intramedullary Guide Group	Spacer Block Group	*p* Value
No. of subjects	39	76	
Age (year)	64.9 ± 9.1	66.4 ± 6.3	n.s †
Sex (male/female)	4/35	6/70	n.s *
BMI (kg/m^2^)	26.0 ± 3.3	26.2 ± 3.2	n.s †
Clinical parameters			
Flexion contracture of knee (°)	2.6 ± 4.0	3.8 ± 6.2	n.s †
Maximal flexion angle of knee (°)	136.7 ± 8.4	132.4 ± 11.4	n.s †
Hospital for Special Surgery score	64.0 ± 12.1	59.7 ± 19.8	n.s †
Knee Society Knee Score	54.1 ± 20.8	48.4 ± 21.0	n.s †
Knee Society Function Score	41.8 ± 17.9	37.6 ± 16.1	n.s †
Pain visual analog scale (0–10)	6.5 ± 2.0	7.2 ± 1.9	n.s †
WOMAC	48.0 ± 12.2	48.5 ± 11.3	n.s †
Radiological parameters			
Mechanical femorotibial angle (°)	−6.6 ± 4.1	−5.3 ± 2.9	0.05 †
Medial proximal tibial angle (°)	−3.5 ± 2.6	−3.3 ± 2.2	n.s †
Tibial posterior slope (°)	9.9 ± 2.8	9.7 ± 3.0	n.s †

BMI, body-mass index; WOMAC, Western Ontario and McMaster Universities Arthritis Index; n.s, not significant. Related to Mechanical femorotibial angle and Medial proximal tibial angle, varus alignment was designated as negative values, and valgus alignment was designated as positive values. * Derived with Pearson chi-square test. † Derived with Student’s *t*-test.

**Table 2 medicina-59-00089-t002:** Comparison of postoperative clinical and radiological data between patients in the intramedullary guide and spacer block groups.

Variable	Intramedullary Guide Group (N = 39)	Spacer Block Group (N = 76)	*p* Value
Clinical parameters			
Flexion contracture of knee (°)	1.2 ± 2.6	0.3 ± 1.1	n.s
Maximal flexion angle of knee (°)	134.4 ± 8.8	135.7 ± 5.5	n.s
Hospital for Special Surgery score	95.6 ± 3.0	90.3 ± 9.0	n.s
Knee Society Knee Score	93.5 ± 4.5	92.5 ± 9.2	n.s
Knee Society Function Score	88.5 ± 8.4	91.7 ± 8.0	n..s
Pain visual analog scale (0–10)	1.0 ± 1.3	1.5 ± 1.7	n.s
WOMAC	8.9 ± 7.0	9.1 ± 13.3	n.s
Radiological parameters			
Mechanical FT angle (°)	−1.8 ± 3.7	−0.3 ± 3.2	0.025
Difference from preoperative value (°)	4.8 ± 3.5	5.0 ± 2.2	n.s
Coronal femoral component angle (α) (°)	2.4 ± 4.9	−1.1 ± 3.2	<0.001
Number of outlier in α angle (<−10 or 10<)	3	0	0.014
Coronal tibial component angle (β) (°)	−1.0 ± 4.3	−3.0 ± 2.4	0.001
Difference from preoperative MPTA (°)	2.4 ± 4.9	0.2 ± 2.8	0.003
Sagittal femoral component angle (°)	3.1 ± 8.7	5.0 ± 4.1	n.s
Sagittal tibial component angle (°)	7.2 ± 3.5	8.2 ± 2.7	n.s
Difference from preoperative tibial posterior slope (°)	−2.8 ± 4.0	−1.4 ± 3.4	n.s
Coronal FT component congruence angle (°)	3.4 ± 4.5	1.9 ± 2.6	0.028

FT, femorotibial; WOMAC, Western Ontario and McMaster Universities Arthritis Index; DFBA, distal femoral bowing angle; MPTA, medial proximal tibial angle; n.s, not significant. In representing mechanical FT angle, coronal femoral.

## Data Availability

The data presented in this study are available on request from the corresponding author. The data are not publicly available due to privacy.

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
