# Peer review of "Spacer Block Technique Was Superior to Intramedullary Guide Technique in Coronal Alignment of Femoral Component after Fixed-Bearing Medial Unicompartmental Knee Arthroplasty: A Case–Control Study"

_medicina, 2022, doi:10.3390/medicina59010089_

Round 1

Reviewer 1 Report

The manuscript is very well written. 

The results are well documented with statistical analysis. 

I suggest that you add a schematic or image of the technique used

Author Response

Point 1: The manuscript is very well written. The results are well documented with statistical analysis. I suggest that you add a schematic or image of the technique used.

Response 1: We added s scheme of the technique as you recommended.

Reviewer 2 Report

This is well executed and focused study. The main concerns are the retrospective nature of the study and the comparison of different implants. Both are addressed sufficiently in the discussion.

Line 30: after range insert comma not a semicolon

In the conclusion: put the last part of the sentence first

Line 32: the femoral

Line 64: surgeon subjectivity and experience are contributing factors. The phrase is unclear. Please be more specific

Line 70: and intends the femur cut parallel. Improve sentence.

Lines 70-73: are better suited on the start of the paragraph

Lines 95-96: describe the statistical test used in every comparison

Line 105: are the surgeon part of this study. In that case refer to them with their initials

The results are very clearly presented. Do you think it is not appropriate to perform a power analysis in order to find the patient population necessary to reveal a significant difference?

The extent of the Discussion could be reduced and be focused primarily on the clinical efficiency of  UKA

Author Response

Point 1: Line 30: after range insert comma not a semicolon.

Response 1: We changed semicolon to comma as you recommended.

Point 2: In the conclusion: put the last part of the sentence first.

Response 2: We moved the last part to the first part of the conclusion as you recommended.

Point 3: Line 32: the femoral

Response 3: I changed ‘femoral’ to ‘the femoral’.

Point 4: Line 64: surgeon subjectivity and experience are contributing factors. The phrase is unclear. Please be more specific

Response 4: We made the sentence more specific by changing some words. We change the sentence to ‘surgeon’s technical proficiency and subjective assessment of gap balancing are contributing factors’.

Point 5: Line 70: and intends the femur cut parallel. Improve sentence.

Response 5: We changed ‘and intends the femur cut parallel’ to ‘parallel to the tibial cutting surface on the coronal plane’.

Point 6: Lines 70-73: are better suited on the start of the paragraph

Response 6: We moved this part on the start of the paragraph as you recommended.

Point 7: Lines 95-96: describe the statistical test used in every comparison

Response 7: We described the statistical test used in every comparison in Table 1.

Point 8: Line 105: are the surgeon part of this study. In that case refer to them with their initials.

Response 8: We adde the surgeon’s initials.

Point 9: Do you think it is not appropriate to perform a power analysis in order to find the patient population necessary to reveal a significant difference?

Response 9: Due to the retrospective nature of the study, an a priori sample size calculation was not possible. Instead, a post-hoc power analysis was applied using a 2-sided test at an alpha level of 0.05 with a power of 80%, in order to determine the sample size for statistical significance with a medium effect size. We added the power analysis to show the adequate sample size.

Point 10: The extent of the Discussion could be reduced and be focused primarily on the clinical efficiency of  UKA.

Response 10: We reduced the discussion section and focused primarily on the clinical efficiency of  UKA as you recommended.
